# Molecular Docking and Biophysical Studies for Antiproliferative Assessment of Synthetic Pyrazolo-Pyrimidinones Tethered with Hydrazide-Hydrazones

**DOI:** 10.3390/ijms22052742

**Published:** 2021-03-08

**Authors:** Mabrouk Horchani, Gerardo Della Sala, Alessia Caso, Federica D’Aria, Germana Esposito, Ilaria Laurenzana, Concetta Giancola, Valeria Costantino, Hichem Ben Jannet, Anis Romdhane

**Affiliations:** 1Laboratory of Heterocyclic Chemistry, Natural Products and Reactivity, Medicinal Chemistry and Natural Products (LR11ES39), Faculty of Sciences of Monastir, University of Monastir, 5000 Monastir, Tunisia; horchani.mabrouk@gmail.com (M.H.); anis_romdhane@yahoo.fr (A.R.); 2Department of Marine Biotechnology, Stazione Zoologica Anton Dohrn, Villa Comunale, 80125 Naples, Italy; gerardo.dellasala@unina.it; 3Laboratory of Pre-Clinical and Translational Research, IRCCS-CROB, Referral Cancer Center of Basilicata, 85028 Rionero in Vulture, Italy; ilaria.laurenzana@crob.it; 4The Blue Chemistry Lab, Department of Pharmacy, University of Naples Federico II, Via D. Montesano 49, 80131 Naples, Italy; alessia.caso@unina.it (A.C.); germana.esposito@unina.it (G.E.); 5Department of Pharmacy, University of Naples Federico II, Via D. Montesano 49, 80131 Naples, Italy; federica.daria@unina.it (F.D.); giancola@unina.it (C.G.)

**Keywords:** pyrazolopyrimidinone, hydrazide-hydrazone, antitumor lead compound, cytotoxic activity, antiproliferative activity, molecular docking, epidermal growth factor receptor, G-quadruplex DNA, KRAS, circular dichroism

## Abstract

Chemotherapy represents the most applied approach to cancer treatment. Owing to the frequent onset of chemoresistance and tumor relapses, there is an urgent need to discover novel and more effective anticancer drugs. In the search for therapeutic alternatives to treat the cancer disease, a series of hybrid pyrazolo[3,4-*d*]pyrimidin-4(*5H*)-ones tethered with hydrazide-hydrazones, **5a**–**h**, was synthesized from condensation reaction of pyrazolopyrimidinone-hydrazide **4** with a series of arylaldehydes in ethanol, in acid catalysis. In vitro assessment of antiproliferative effects against MCF-7 breast cancer cells, unveiled that **5a**, **5e**, **5g**, and **5h** were the most effective compounds of the series and exerted their cytotoxic activity through apoptosis induction and G0/G1 phase cell-cycle arrest. To explore their mechanism at a molecular level, **5a**, **5e**, **5g**, and **5h** were evaluated for their binding interactions with two well-known anticancer targets, namely the epidermal growth factor receptor (EGFR) and the G-quadruplex DNA structures. Molecular docking simulations highlighted high binding affinity of **5a**, **5e**, **5g**, and **5h** towards EGFR. Circular dichroism (CD) experiments suggested **5a** as a stabilizer agent of the G-quadruplex from the *Kirsten ras* (KRAS) oncogene promoter. In the light of these findings, we propose the pyrazolo-pyrimidinone scaffold bearing a hydrazide-hydrazone moiety as a lead skeleton for designing novel anticancer compounds.

## 1. Introduction

Cancer is a main health issue worldwide due to the yearly increases in the number of patients with this real disease [1]. According to the World Health Organization (WHO), cancer cases, which were discovered in 2012, surpassed 14 million, and it is expected to reach 22 million in 2030 [2]. Discovery of new anticancer drug candidates is extremely challenging, mainly due to the metabolic heterogeneity and molecular complexity of tumors [3], as well as to the onset of chemoresistance mechanisms and tumor relapses. In the last few years, faced with the urgent need for new drug launches, our research group has explored both natural products inspired synthesis and chemical modification of synthetic small molecules for antitumor lead identification [4,5]. Although natural products are a reservoir of bioactive scaffolds [6,7,8,9,10,11], synthesis of small molecules represents a powerful and effective strategy to provide novel chemical entities for cancer therapy, primarily for their easy availability. Notably, fragment-based lead discovery approaches together with building block analysis of approved drugs has provided useful hints in designing drug-like compounds in medicinal chemistry [12,13]. In addition, integration of cell-based and/or cell-free bioassays and in silico molecular docking allows, at an early stage, filtering out of molecules lacking any biological effects.

In this light, and as known, heterocyclic compounds occupy a pivotal position in modern medicinal chemistry. More especially, nitrogenous cyclic moieties play a vital role in the synthesis of potent biomolecules.

Indeed, pyrazoles are reported to exhibit cytotoxic [14], anti-inflammatory [15], anti-HCV, and antitumor activity [16]. Furthermore, pyrimidinone derivatives constitute a promising versatile class of heterocyclic compounds, which recently attracted much interest from chemists due to their outstanding potential, such as insecticide [17], anti-tyrosinase [18], antimicrobial [19], and anticancer agents [20]. Pyrazole-fused pyrimidinones are considered as the most ubiquitous heterocyclic privileged scaffolds and these moieties are present in a variety of potent therapeutic compounds with different pharmacological properties, such as antioxidant [21], antituberculosis [22], cardiovascular [23], antimicrobial [24], anticancer [25], and cytotoxic activities (Figure 1) [26,27,28].

On another hand, the pharmacophore ingredient with CO-NH-N=CH (hydrazide-hydrazones) has received considerable attention due to its broad spectrum of biological applications. Thus, hydrazide-hydrazones are known as anti-diabetic [29], antileishmanial [30], antimalarial [31], and anti-tumor compounds [32]. Hydrazide-hydrazones linked to heterocyclic moiety as pyrazole or pyrimidine are known to have a significant cytotoxic potential, as in the case of molecules D and E in Figure 1 [33,34].

In view of the above-mentioned findings, and in the search for therapeutic alternatives to treat the cancer disease, we designed the synthesis of novel hybrid conjugates through combination of two distinct cytotoxic pharmacophores, namely the pyrazolo-pyrimidinone and the hydrazide-hydrazone motifs. A small library of twelve molecules was generated and evaluated for its anticancer potential in the human breast adenocarcinoma MCF-7 cell line, to select the most potent lead compounds for mechanism-of-action and target-identification studies. MCF-7 cells were used as a model in our screening, as breast cancer, together with lung cancer, was the most common cancer worldwide in 2018, according to WHO estimates [2].

Particularly, in a lead optimization perspective, this novel pyrazolo-pyrimidinone scaffold bearing a hydrazide-hydrazone moiety, was probed for its ability to (1) interact with the epidermal growth factor receptor (EGFR) tyrosine kinase and (2) stabilize G-quadruplex DNA structures, currently known as anticancer targets for cancer therapy [35,36,37].

EGFR is a membrane receptor overexpressed in many solid tumors, including breast, ovarian, head-and-neck, renal, prostate, colon, pancreas, and non-small-cell lung cancer [38]. EGFR is a member of the ErbB family (originally named because of their homology to the erythroblastoma viral gene product, *v-erbB*) of structurally related receptor tyrosine kinases (RTKs) and controls downstream signaling pathways, inducing cell growth, cell cycle progression, angiogenesis, cell motility and apoptosis inhibition. Tumors showing activating mutations in the EGFR gene are strikingly dependent on EGFR activity for their growth and survival [39]. Therefore, several agents specifically addressed to the EGFR have been developed, such as the monoclonal antibody cetuximab and three generations of tyrosine kinase inhibitors (e.g., erlotinib, afatinib, and osimertinib).

The G-quadruplex (G4) is a four-stranded structure formed by G-rich sequences, either DNA or RNA, which has at least two stacked G-tetrads [40,41]. A G-tetrad is a planar square arrangement of four guanines stabilized by Hoogsteen hydrogen bonds and monovalent cations [42]. During the last decade, G4s have emerged as possible druggable anticancer targets because they are located in key genome regions, like telomeres and oncogene promoters. In particular, small molecules that bind and stabilize the G4 in oncogene promoters may lead to downregulation of oncogene expression [43]. In this work, we focused our attention on the G4 structures within *Kirsten ras* (*KRAS*) and *B-cell lymphoma-2* (*BCL2*) genes. The *KRAS* gene is overexpressed in about 30% of all human cancers [44]. The Bcl-2 family of proteins is the main regulator of apoptotic process, acting either to promote or inhibit it [45]. Overexpression of *BCL2* gene, which encodes the anti-apoptotic Bcl-2 protein, greatly contributes to the resistance of cancer cells to apoptosis, and it has also been reported to play a role in the resistance to conventional cancer treatments [46]. Although significant progress has been made in this research area, with some compounds reaching the advanced phases of clinical trials [47], a therapeutic approach based upon G4 ligands is challenging as many of them display low selectivity to the targeted G4 structure. Therefore, the search for novel chemical entities targeting G quadruplexes is of huge interest.

## 2. Results

### 2.1. Chemistry

The 5-amino-3-methyl-1-phenyl-*1H*-pyrazole-4-carbonitrile **1** has served as a key starting material. It was prepared according to the previously reported method [24]. In order to synthesize the target pyrazolopyrimidinone bearing hydrazide-hydrazone derivatives, the hydrazide **3** has been prepared as depicted in Scheme 1. The first step involves the synthesis of pyrazolopyrimidinone **2** by reaction of precursor **1** under reflux of acetic anhydride and in the presence of a determined volume of phosphoric acid. Compound **2** was treated with ethyl chloroacetate affording compound **3** in 75% yield. Then, ester **3** was submitted to nucleophilic substitution with hydrazine monohydrate obtaining pyrazolo-pyrimidinic hydrazide **4** in 55% yield. Structures of compounds **1**, **2**, **3**, and **4** were established through their ^1^H NMR and ^13^C NMR spectra after purification and the results are shown in the experimental section. Subsequently, the key **4** was coupled with different arylaldehydes in ethanol and some drops of hydrochloric acid, thus obtaining hydrazide-hydrazones **5a**–**h** in 60–78% yields (Table 1). The synthetic strategy is depicted in Scheme 2. The structures of the target derivatives were characterized by ^1^H NMR, ^13^C NMR, and high resolution mass spectra (HRMS) spectroscopic techniques. Indeed, the ^1^HNMR spectra of compounds **5** showed the disappearance of the signals at 4.34 ppm relative to « –NH_2_ » of hydrazide and the appearance of new signals corresponding to the protons introduced by arylaldehyde. Analysis of ^13^C NMR spectra of the same compounds demonstrated the desired structures through the appearance of aromatic and the iminic carbons C_5′._ Furthermore, the electrospray ionization high resolution mass spectra (ESI-HRMS) of all the examined compounds **5a–h** showed the correct protonated molecular ion peaks [M+H]^+^, which is consistent with the molecular formula.

### 2.2. In Vitro Evaluation of Antiproliferative Activity in MCF-7 Breast Cancer Cells

Twelve synthetic compounds (**1**, **2**, **3**, **4**, **5a**, **5b**, **5c**, **5d**, **5e**, **5f**, **5g**, and **5h**) were initially assessed for their antitumor activity against breast adenocarcinoma MCF-7 cell line, at a single dose exposure (50 µM). Cancer cell viability was monitored for 48 h after drug administration, by using the real-time xCELLigence cell analyzer, which allows to evaluate proliferation of cells growing on microsensor electrodes. Drug-induced antiproliferative effects elicit electronic impedance reduction, which is translated into a unit-less parameter, namely cell index (CI), thereby reflecting cellular adhesion, growth, and morphology states.

Real time monitoring of cancer cell proliferation unveiled that **5a**, **5e**, **5g**, and **5h** were the most bioactive molecules of the series and were shown to exert significant antiproliferative activity already within the first 24 h of treatment (Figure 2). Notably, MCF-7 cells experienced a substantial drop in the CI value after incubation with 50 µM of compound **5h**, which resulted to be approximately 2 times more potent than cisplatin, one of the most widely used and effective chemotherapeutic drugs for treatment of solid tumors, such as ovarian, testicular, and bladder carcinoma [48]. Moreover, cisplatin is also used as adjuvant/neoadjuvant agent in breast cancer patients and as a single chemo drug or in combination with other medications, for the treatment of advanced and triple negative breast cancer.

Growth inhibition was rather severe but less pronounced in MCF-7 cells exposed to **5a**, **5e**, and **5g** (50 µM)**,** with the latter exhibiting similar extent of cytotoxicity as cisplatin towards breast cancer cells (Figure 2A).

In single treatment, MCF-7 cells were incubated with different concentrations (6.25, 12.5, 25, 50, 100 µM) of **5a**, **5e**, **5g**, and **5h** to assess the half-maximal inhibitory concentration (IC_50_) of individual synthetic derivatives. The growth rate of tumor cells, monitored by xCELLigence real-time cell analysis, was significantly inhibited by each of the aforementioned molecules, in a dose-dependent manner. The IC_50_ values of **5a**, **5e**, **5g**, and **5h**, determined after 24 h of incubation, were 55.3, 60.0, 45.4, and 34.6 µM, respectively (Figure 2B).

To test whether apoptosis induction contributed to growth inhibitory potency of **5a**, **5e**, **5g**, and **5h**, we evaluated the apoptotic rate of MCF-7 cells by an Annexin V-fluorescein isothiocyanate (FITC)/Propidium iodide (PI) staining assay (Figure 3). MCF-7 cells were treated with IC_50_ concentrations of each compound for 24 h. All compounds were shown to induce significant apoptotic cell death in breast cancer cells as compared to controls exposed to 0.5% dimethyl sulfoxide (DMSO) vehicle, with **5e** and **5g** causing higher rates of apoptosis (69.6% and 59.3%, respectively). Interestingly, although apoptosis was the major cell death mechanism, **5h** turned out to be the only compound to trigger mild but significant necrotic cell injury.

Aiming to assess whether cell cycle arrest could underly the antiproliferative effects of **5a**, **5e**, **5g**, and **5h** in MCF-7 cancer cells, flow cytometry cell cycle analysis was performed using propidium iodide DNA staining in live cells (Figure 4). Incubating MCF-7 with IC_50_ concentrations of each compound for 24 h led to cell cycle arrest in the G_1_/G_0_ phase, thereby decreasing the proportion of cells in the S and G2/M phases.

Next, we evaluated detailed response curves for compounds **5a**, **5e**, **5g**, and **5h** in normal human dermal fibroblasts (NHDF) to probe putative differential effects on tumor versus normal cells. NHDF were exposed to increasing doses of **5a**, **5e**, **5g**, and **5h** (6.25, 12.5, 25, 50, 100 µM) and then, cell viability was monitored for 48 h, using the xCELLigence system (Appendix A). The cytotoxic effects of the four synthesized molecules in breast cancer cells were not selective as they also inhibited cell survival in the normal cell model, with almost similar IC_50_ values. However, **5a**, **5e**, **5g**, and even the most potent compound, **5h**, were shown to exert less toxicity than cisplatin against NHDF cells (Appendix A).

### 2.3. Identification of Potential Anticancer Drug Targets of **5a**, **5e**, **5g**, and **5h**

#### 2.3.1. In Silico Molecular Docking of **5a**, **5e**, **5g**, and **5h** as Epidermal Growth Factor Receptor Tyrosine Kinase Inhibitors

EGFR is the first member of the Her receptor family which plays a conspicuous role in cellular signaling activities, including cell proliferation, growth, adhesion, differentiation, metabolism, motility, and death [49,50,51]. Therefore, EGFR has been regarded as a valuable anticancer target in medicinal chemistry. As several EGFR inhibitors shared structural moieties with our synthetic compounds [52,53,54], we explored putative binding interactions of **5a, 5e, 5g**, and **5h** with this receptor. In addition, EGFR inhibition has been already demonstrated to enhance cell death pathways (e.g., apoptosis, autophagy) in MCF7 cells [55,56,57].

With this regard, a literature survey revealed that molecular docking into the ATP binding site of EGFR (protein data bank ID: 1M17) was a robust strategy for identification of effective EGFR inhibitors endowed with cytotoxic properties [58,59,60]. In the current study, docking simulations using the crystal structure of EGFR, co-crystallized with its inhibitor erlotinib, has been performed to elucidate the interactions of the active cytotoxic agents **5a, 5e, 5g**, and **5h** at the ‘Erlotinib’ binding domain of tyrosine kinase enzyme (protein data bank ID: 1M17), by using Autodock 4.2 (Table 2). The EGFR protein was previously pre-treated by removing all bound water molecules and adding the polar hydrogen atoms and Gasteiger charges to the system during the preparation of the receptor input file. Then, AutoDock Tools were used for the preparation of the corresponding ligand and protein files (PDBQT). Next, pre-calculation of the grid maps was performed using Auto Grid for saving time during the docking procedure.

The majority of cytotoxic agents reported so far, resembles the binding interaction of Erlotinib within the active site.

Indeed, regarding to the hydrazide-hydrazone conjugates **5a, 5e, 5g**, and **5h**, it can be noticed that they are involved in conventional Hydrogen bond with MET-A-769 through NH group. Pi-Sulfur bond was observed between MET-A-742 (highlighted in yellow color in Figure 5) and the phenyl moiety. Furthermore, Pi-Anion bond was formed by the pyrazole ring and ASP-A-831 (highlighted in golden color in Figure 5), in addition to Pi-Sigma interactions formed by the pyrazolopyrimidinone fragment and amino acids: VAL-A-702 and LEU-A-820. Moreover, all derivatives showed many hydrophobic interactions (Alkyl and Pi-Alkyl with amino acids sequence: LEU-A-694, VAL-A-702, LYS-A-704, ALA-A719, LYS-A-721, MET-A-769, and LEU-A-820), except for **5g** (having the lowest binding energy values), which displayed a second hydrogen bond through its hydroxy group with PRO-A-770. This finding demonstrates that this compound is more active than its analogs. On another hand, the conjugate **5h** is nicely bound to the ATP binding site of EGFR through an Alkyl interaction with PHE-A-699, in addition to a Carbon-Hydrogen bond formed by the CH group of hydrazide-hydrazone pharmacophore with MET-A-769, as well as the other interactions mentioned above (Figure 5). From this binding model, it could be concluded that these last interactions are responsible for the effective EGFR inhibitory of these compounds, especially **5g** and **5h**.

#### 2.3.2. Physicochemical Evaluation of **5a**, **5e**, **5g**, and **5h** as G-Quadruplex DNA Stabilizers

To investigate the in vitro binding properties of synthesized molecules versus G4 structures Circular Dichroism (CD) spectroscopy and CD melting experiments were carried out. Two G4 forming sequences from *KRAS* (*KRAS* 22RT) and *BCL2* (*BCL2*-G4) oncogene promoter regions, and a G4 from human telomere (Tel23), were used in these experiments. A 20-mer hairpin-duplex DNA consisting of two self-complementary 8-mer sequences connected by a TTTT loop (hairpin duplex) was also used to estimate the G4 over duplex selectivity of the ligands. Initially, the conformation of each DNA sample was verified by CD spectroscopy. The CD spectrum of *KRAS* 22RT, showed a positive band at 264 nm and a negative band at 240 nm, in agreement with the presence of parallel G4 topology (Figure 6). Otherwise, BCL2-G4 and Tel 23 showed two positive bands at around 265 and 290 nm and a weak negative band at 240 nm, in agreement with a hybrid structure as major conformation. The structure of hairpin duplex was also verified by CD, showing the typical spectrum of a duplex DNA (Appendix A). To verify if the compounds **5a**, **5e**, **5g**, and **5h** alter the native folding topologies of the investigated sequence, CD experiments were performed by adding an excess of each ligand to the pre-folded G4 structure. Notably, no CD spectra changes were observed upon addition of each ligand (Appendix A). The stabilizing effect of each compound was evaluated by CD melting experiments by measuring the ligand-induced change in the melting temperature (Δ*T*_m_) of both G4 and duplex structures. CD melting results show that only **5a** has the capability to stabilize *KRAS* 22RT with a Δ*T*_m_ of +7 °C (Table 3) and shows selectivity for G4 structures over duplex DNA (Figure 6).

## 3. Discussion

In the current work, we have successfully designed and synthesized a new series of pyrazolo-pyrimidinones bearing hydrazide-hydrazone derivatives **5a**–**h** via reaction of hydrazide 4, previously prepared starting from the 5-amino-3-methyl-1-phenyl-*1H*-pyrazole-4-carbonitrile 1 with substituted arylaldehydes. Among these compounds, **5a**, **5e**, **5g**, and **5h** were shown to exert significant cell-growth inhibition in MCF-7 breast cancer cells, thereby inducing apoptosis and G0/G1 phase cell cycle arrest. Notably, in cell viability assays, **5h** was the most bioactive derivative and showed a) more effective antiproliferative properties against MCF-7 cells and b) less cytotoxicity towards normal dermal fibroblasts than cisplatin, which was used as reference drug in our screening. Compounds 5a, **5e**, **5g**, and **5h** did not display selectivity towards cancerous cells as they also inhibited cell survival in the normal cell model, with almost similar IC_50_ values (Figure 2 and Appendix A). However, NHDF cells appeared to be more vulnerable to cisplatin rather than synthetic compounds, which could exhibit higher tolerability and cause less damage to healthy tissues.

The proposed pyrazolo-pyrimidinone scaffold tethered with hydrazide-hydrazone derivatives represents an essential feature to keep cytotoxic activity against MCF-7 cell line. Indeed, compounds 1-4, which lack the aryl-hydrazone moiety, resulted to be among the less bioactive compounds tested in this study. In regard to the chemical nature of substituents on the aryl-hydrazone moiety, two bioactivity trends seemed to emerge from antiproliferative assays: (a) insertion of a bulky halogen atom (such as bromine) in **5h** led to a significant enhancement of cytotoxic activity as compared to the fluorine derivative **5f**; (b) increasing hydrophobic steric hindrance of functional groups, such as -N(CH_3_)_2_ in **5a**, improves antiproliferative effects with respect to the methoxy derivative **5b**.

Interestingly, the greater cytotoxicity exerted by **5a** and **5h** correlates with their ability to form stronger and more stable interactions with the erlotinib binding site of EGFR, as compared to **5b** and **5f** in molecular docking simulations. Particularly, while **5a** and **5h** reorient the NH group of the hydrazone fragment to form the hydrogen bond with MET-A-769, this interaction is missing for their respective analogues **5b** and **5f** (Figure 7). This is explained by the crucial role of large groups in **5a** and **5h** (N(CH_3_)_2_ and Br, respectively) for the stability of the enzyme-ligand complex and, therefore, for the inhibition of EGFR.

Molecular docking simulations using the crystal structure of EGFR, co-crystallized with its inhibitor erlotinib, provided evidence for high binding affinity of **5a**, **5e**, **5g**, and **5h** towards EGFR. These results suggest that these molecules, at least partially, may exert their cytotoxic effects through EGFR inhibition, which correlates with the observed apoptosis induction and G_0_/G_1_ cell cycle arrest in cancer cells, after exposure to compounds **5a**, **5e**, **5g**, and **5h** [61,62]. The cytotoxic power of these four compounds is explained by the high number of various interactions formed: hydrophobic (Pi-Sigma, Pi-Alkyl, Alkyl), electrostatic (Pi-Anion), Pi-Sulfur, and especially hydrogen bond (Figure 5).

Moreover, physicochemical evaluation of **5a**, **5e**, **5g**, and **5h** as G-Quadruplex DNA stabilizers, showed that only **5a** enhanced thermal stability of *KRAS* 22RT, unveiling selectivity for G4 structures over duplex DNA. Besides the large aromatic flat surface, which is shared in all compounds, **5a** is the only derivative to display a further structural requirement for efficient G4 ligands [63], that is the dimethylamino group at the para-position of the aryl-hydrazone motif. This group may be positively charged at the physiological pH and allow **5a** to establish key electrostatic interactions for binding G4–DNA structures. In addition, CD experiments unveiled **5e** and **5h** to bind hairpin duplex DNA, thereby suggesting these molecules could form DNA adducts, contributing to the cytotoxic mechanism enhanced by these molecules.

Overall, our findings clearly indicate that the pyrazolo-pyrimidinone scaffold tethered with hydrazide-hydrazone derivatives represents a promising cytotoxic pharmacophore for anticancer drug development. Therefore, our efforts will be addressed toward improvement of biological properties of this chemical backbone, aiming to design either (a) selective EGFR inhibitors, using **5g** and **5h** as pivotal molecules or (b) more efficient G4 ligands, using **5a** as lead compound.

## 4. Materials and Methods

### 4.1. General Information

The control of all reactions was monitored by TLC using aluminium sheets of Merck silica gel 60 F254, 0.2 mm. Melting points were determined on an Electrothermal 9002 melting point apparatus and are uncorrected. NMR spectra were recorded on a Bruker Avance Neo spectrometer (Bruker BioSpin Corporation, Billerica, MA, USA) at 400 MHz (^1^H) and 100 MHz (^13^C) using dimethylsulfoxide-*d*_6_ as solvent. All chemical shifts were reported as δ values (ppm) relative to residual solvent signal (δ_H_ 2.50, δ_C_ 39.5). High Resolution Mass Spectra (HRES-MS) were obtained with Thermo LTQ Orbitrap XL mass spectrometer (Thermo Fisher Scientific Inc., Waltham, MA, USA) combined to a Thermo U3000 high-performance liquid chromatography (HPLC) system (ESI technique, positive mode). The starting material **1** was prepared according to the literature [64].

### 4.2. Chemistry

#### 4.2.1. General Procedure for the Synthesis of 3,6-dimethyl-1-phenyl-1,5-dihydro-4H-pyrazolo [3,4-d]pyrimidin-4-one 2

In a 100 mL three-necked flask, 4mmol of aminopyrazole **1** is introduced in 20 mL of acetic anhydride. Then 10 mL of phosphoric acid is added dropwise. The mixture is then brought to reflux for 3 h with magnetic stirring. The progress of the reaction is monitored by TLC (eluent 2: 8 ethyl acetate/chloroform) which showed the appearance of a new stain more polar than that of aminopyrazole **1**. After cooling, ice water is added to the reaction crude, the precipitated solid was filtered, washed with cold ether, dried, and then recrystallized from ethanol to give pyrimidinone **2**.

Yield: 79%, mp: 277–279 °C. ES-HRMS [M+H]^+^ calcd. for (C_13_H_13_N_4_O)^+^: 241.1084, found: 241.1083. ^1^H NMR (400 MHz, DMSO-*d_6_*): δ (ppm) = 2.39 (s, 3H, H_3_-12), 2.50 (s, 3H, H_3_-13), 7.34–8.03 (m, 5H, H-Ar), 12.22 (brs, 1H, NH). ^13^C NMR (100 MHz, DMSO-*d_6_*): δ (ppm) = 13.3, 21.5, 103.7, 121.3, 126.4, 129.1, 138.4, 145.7, 152.9, 158.59, 158.65.

#### 4.2.2. General Procedure for the Synthesis of Ethyl 2-(3,6-dimethyl-4-oxo-1-phenyl-1,4-dihydro-5H-pyrazolo[3,4-d]pyrimidin-5-yl)acetate 3

Equimolar solution 1 mmol of pyrimidinone **2**, anhydrous potassium carbonate and ethyl chloroacetate was refluxed in dry dimethylformamide (DMF) (60 mL) with continuous stirring for 6 h, into a three-necked flask of 100 mL. Once the starting material is gone, the reaction mixture was then cooled and poured into cold water. Then, the formed precipitate was filtered off, washed with water, dried, and recrystallized from ethanol to give the ester **3**.

Yield: 75%, mp: 156–158 °C. ES-HRMS [M+H]^+^ calcd. for (C_17_H_19_N_4_O_3_)^+^: 327.1452, found: 327.1450. ^1^H NMR (400 MHz, DMSO-*d*_6_): δ (ppm) = 1.22 (t, 3H, *J* = 7 Hz, H_3_-15), 2.58 (s, 3H, H_3_-12), 2.62 (s, 3H, H_3_-13), 4.19 (q, 2H, *J* = 6.96 Hz, H_2_-14), 5.16 (s, 2H, H_2_-1′), 7.32–8.15 (m, 5H, H-Ar). ^13^C NMR (100 MHz, DMSO-*d*_6_): δ (ppm) = 13.90, 14.03, 25.9, 60.8, 62.8, 100.6, 120.6, 126.2, 129.2, 138.5, 142.4, 155.8, 162.5, 165.4, 167.8.

#### 4.2.3. General Procedure for the Synthesis of 2-(3,6-dimethyl-4-oxo-1-phenyl-1,4-dihydro-5H-pyrazolo[3,4-d]pyrimidin-5-yl)acetohydrazide 4

The previously synthesized ester **3** was treated with an excess of hydrazine hydrate in ethanol at room temperature for 1–2 h until a white precipitate formed. The solid obtained was filtered, washed with ethanol, and dried to obtain compound **4**.

Yield: 65%, mp: 166–168 °C. ES-HRMS [M+H]^+^ calcd. for (C_15_H_17_N_6_O_2_)^+^: 313.1408, found: 313.1406. ^1^H NMR (400 MHz, DMSO-*d_6_*, mixture of isomers): δ (ppm) = 2.51 (s, 3H, H_3_-12), 2.53 (s, 2.6H, H_3_-13), 2.59 (s, 0.4H, H_3_-13), 4.34 (s, 2H, -NH_2_), 4.72 (s, 2H, H-1′), 7.27–8.17 (m, 5H, H-Ar), 9.44 (s, 1H, NH). ^13^C NMR (100 MHz, DMSO-*d_6_*): δ (ppm) = 13.2, 23.47, 23.59, 44.2, 102.9, 120.3, 121.1, 126.5, 128.9, 129.1, 138.3, 145.9, 150.8, 157.7, 159.9, 166.2.

#### 4.2.4. General Procedure for the Synthesis of Derivatives **5a**–**h**

An equimolar solution (1 mmol) of hydrazide **4** and arylaldehyde in the presence of a few drops of hydrochloric acid was heated under reflux of ethanol (15 mL) for 9 h. The reaction mixture was cooled to room temperature and the solvent was removed in vacuo and the obtained residue was purified by silica gel chromatography (petroleum ether/ethyl acetate, 65:35) to give compounds **5a**–**h**.

As displaying an amide and an imine functions within the same structure, compounds **5a**–**h** may exist as (a) *E*/*Z* stereoisomers about the C=N bond of hydrazone moiety and (b) *cis*/*trans* amide bond conformers. As N-acyl substituted hydrazones are present in solution mainly as the *E* stereoisomer [65,66], the observed duplication of some ^1^H and ^13^C signals in NMR spectra of compounds **5a**–**h** (e.g., the methylene group at C-1′ and the imine function at C-5′) can be generally attributed to the presence of *cis*/*trans* amide conformers [56,57]. Only compound **5c** appeared to be present as a mixture of *E*/*Z* stereoisomers and *cis*/*trans* amide conformers, according to NMR data.


*(E)-2-(3,6-dimethyl-4-oxo-1-phenyl-1,4-dihydro-5H-pyrazolo[3,4-d]pyrimidin-5-yl)-N’-(4-(dimethylamino)benzylidene)acetohydrazide*
**5a**


Yield: 60%, mp: 258–260 °C. ES-HRMS [M+H]^+^ calcd. for (C_24_H_26_N_7_O_2_)^+^: 444.2142, found: 444.2139. ^1^H NMR (400 MHz, DMSO-*d_6_*, mixture of isomers): δ (ppm) = 2.52 (s, 3H, H_3_-12), 2.56 (s, 2.4H, H_3_-13), 2.59 (s, 0.6H, H_3_-13), 2.97 (s, 1.8H, H_3_-a/b), 2.98 (s, 4.2H, H_3_-a/b), 4.85 (s, 0.4H, H_2_-1′), 5.28 (s, 1.6H, H_2_-1′), 6.74–8.08 (m, 9H, H-Ar), 7.94 (s, 0.7H, H-5′), 8.08 (s, 0.3H, H-5′), 11.56 (s, 0.9H, NH), 12.34 (s, 0.1H, NH). ^13^C NMR (100 MHz, DMSO-*d*_6_): δ (ppm) = 13.2, 13.3, 23.58, 23.76, 40.8, 45.6, 102.9, 111.8, 121.15, 121.26, 121.3, 126.6, 128.3, 129.1, 129.2, 138.3, 145.3, 145.9, 150.9, 151.5, 157.8, 158.7, 160.0, 167.5.


*(E)-2-(3,6-dimethyl-4-oxo-1-phenyl-1,4-dihydro-5H-pyrazolo[3,4-d]pyrimidin-5-yl)-N’-(4-methoxybenzylidene)acetohydrazide*
**5b**


Yield: 75%, mp: 246–248 °C. ES-HRMS [M+H]^+^ calcd. for (C_23_H_23_N_6_O_3_)^+^: 431.1826, found: 431.1826 ^1^H NMR (400 MHz, DMSO-*d_6_*, mixture of isomers): δ (ppm) = 2.52 (s, 3H, H_3_-12), 2.56 (s, 2.6H, H_3_-13), 2.59 (s, 0.4H, H_3_-13), 3.80 (s, 0.8H, H_3_-14), 3.81 (s, 2.2H, H_3_-14), 4.88 (s, 0.5H, H_2_-1′), 5.30 (s, 1.5H, H_2_-1′), 7.02–8.06 (m, 9H, H-Ar), 8.06 (s, 1H, H-5′), 11.77 (s, 1H, NH). ^13^C NMR (100 MHz, DMSO-*d_6_*): δ (ppm) = 13.69, 13.81, 24.0, 24.2 45.0, 55.8, 103.3, 114.8, 121.7, 127.0, 129.0, 129.2, 129.5, 127.6, 138.7, 144.8, 146.4, 151.4, 153.4, 158.3, 160.5, 161.3, 168.3.


*(E)-2-(3,6-dimethyl-4-oxo-1-phenyl-1,4-dihydro-5H-pyrazolo[3,4-d]pyrimidin-5-yl)-N’-(4-nitrobenzylidene)acetohydrazide*
**5c**


Yield: 78%, mp: 244–246 °C. ES-HRMS [M+H]^+^ calcd. for (C_22_H_20_N_7_O_4_)^+^: 446.1571, found: 446.1567. ^1^H NMR (400 MHz, DMSO-*d_6_*, mixture of isomers): δ (ppm) = 1.91 (s, 2.2H, H_3_-12), 1.93 (s, 0.8H, H_3_-12), 1.95 (s, 0.7H, H_3_-13), 1.99 (s, 2.3H, H_3_-13), 2.52 (s, 3H, H_3_-12), 2.56 (s, 2.6H, H_3_-13), 2.59 (s, 0.4H, H_3_-13), 4.90 (s, 0.5H, H_2_-1′), 4.93 (s, 0.3H, H_2_-1′), 5.15 (s, 1.5H, H_2_-1′), 5.36 (s, 0.8H, H_2_-1′), 7.33–8.29 (m, 18H, H-Ar), 8.3 (s, 0.5H, H-5′), 8.5 (s, 0.5H, H-5′), 10.52 (s, 0.2H, NH), 10.66 (s, 0.8H, NH), 12.16 (s, 0.2H, NH), 12.23 (s, 0.8H, NH). ^13^C NMR (100 MHz, DMSO-*d_6_*): δ (ppm) = 13.2, 13.3, 23.48 23.56, 44.8, 102.9, 121.2, 124.0, 126.4, 127.9, 129.0, 129.1, 138.3, 145.6, 150.8, 152.2, 157.8, 158.7, 160.0, 168.2.


*(E)-N’-(3,4-dimethoxybenzylidene)-2-(3,6-dimethyl-4-oxo-1-phenyl-1,4-dihydro-5H-pyrazolo[3,4-d]pyrimidin-5-yl)acetohydrazide*
**5d**


Yield: 70%, mp: 255–257 °C. ES-HRMS [M+H]^+^ calcd. for (C_24_H_25_N_6_O_4_)^+^: 461.1932, found: 461.1929. ^1^H NMR (400 MHz, DMSO-*d*_6_, mixture of isomers): (ppm) = 2.52 (s, 3H, H_3_-12), 2.56 (s, 2.2H, H_3_-13), 2.59 (s, 0.8H, H_3_-13), 3.80 (s, 1H, H_3_-14/15), 3.82 (s, 5H, H_3_-14/15), 4.88 (s, 0.5H, H_2_-1′), 5.32 (s, 1.5H, H_2_-1′), 7.02–7.08 (m, 8H, H-Ar), 7.99 (s, 0.8H, H-5′), 8.16 (s, 0.2H, H-5′), 11.75 (s, 1H, NH). ^13^C NMR (100 MHz, DMSO-*d*_6_): δ (ppm) = 13.2, 13.3, 23.6, 44.6, 45.1, 55.4, 55.5, 102.8, 108.6, 111.5, 121.2, 121.4, 126.5, 129.0, 129.1, 138.2, 144.5, 145.9, 149.0, 150.7, 150.9, 157.8, 159.9, 167.9.


*(E)-2-(3,6-dimethyl-4-oxo-1-phenyl-1,4-dihydro-5H-pyrazolo[3,4-d]pyrimidin-5-yl)-N’-(3-ethoxy-4-hydroxybenzylidene)acetohydrazide*
**5e**


Yield: 62%, mp: 242–244 °C. ES-HRMS [M+H]^+^ calcd. for (C_24_H_25_N_6_O_4_)^+^: 461.1932, found: 461.1926. ^1^H NMR (400 MHz, DMSO-*d_6_*, mixture of isomers): δ (ppm) = 1.35 (t, *J* = 6.95 Hz, 3H, H_3_-15), 2.52 (s, 3H, H_3_-12), 2.56 (s, 2.3H, H_3_-13), 2.59 (s, 0.7H, H_3_-13), 4.08 (q, *J* = 6.92 Hz, 2H, H_2_-14), 4.9 (s, 0.5H, H_2_-1′), 5.29 (s, 1.5H, H_2_-1′), 6.85–8.06 (m, 8H, H-Ar), 7.94 (s, 0.8H, H-5′), 8.06 (s, 0.2H, H-5′), 9.48 (s, 1H, OH), 11.67 (s, 1H, NH). ^13^C NMR (100 MHz, DMSO-*d_6_*): δ (ppm) = 13.2, 14.7, 23.57, 23.71, 44.6, 63.84, 63.89, 102.9, 110.6, 110.9, 115.6, 121.2, 121.5, 121.9, 125.2, 126.5, 129.2, 138.2, 144.9, 145.3, 145.9, 147.1, 147.8, 149.2, 150.9, 157.77, 157.83, 160.0, 162.9, 167.8.


*(E)-2-(3,6-dimethyl-4-oxo-1-phenyl-1,4-dihydro-5H-pyrazolo[3,4-d]pyrimidin-5-yl)-N’-(4-fluorobenzylidene)acetohydrazide*
**5f**


Yield: 68%, mp: 250–252°C. ES-HRMS [M+H]^+^ calcd. for (C_22_H_20_FN_6_O_2_)^+^: 419.1626, found: 419.1626. ^1^H NMR (400 MHz, DMSO-*d_6_*, mixture of isomers): δ (ppm) = 2.52 (s, 3H, H_3_-12), 2.56 (s, 2.5H, H_3_-13), 2.59 (s, 0.5H, H_3_-13), 4.89 (s, 0.4H, H_2_-1′), 5.32 (s, 1.6H, H_2_-1′), 7.27–8.24 (m, 9H, H-Ar), 8.08 (s, 1H, H-5′), 11.89 (s, 1H, NH). ^13^C NMR (100 MHz, DMSO-*d*_6_): δ (ppm) = 13.68, 24.0, 45.0, 103.3, 104.2, 116.3, 116.5, 121.8, 127.1, 129.7, 131.0, 138.7, 138.9, 143.8, 146.4, 151.4, 158.3, 160.4, 168.7.


*(E)-N’-(5-bromo-2-hydroxybenzylidene)-2-(3,6-dimethyl-4-oxo-1-phenyl-1,4-dihydro-5H-pyrazolo[3,4-d]pyrimidin-5-yl)acetohydrazide*
**5g**


Yield: 64%, mp: 253–255 °C. ES-HRMS [M+H]^+^ calcd. for (C_22_H_20_BrN_6_O_3_)^+^: 495.0775, found: 495.0768. ^1^H NMR (400 MHz, DMSO-*d*_6_, mixture of isomers): δ (ppm) = 2.52 (s, 3H, H_3_-12), 2.56 (s, 2.2H, H_3_-13), 2.60 (s, 0.8H, H_3_-13), 4.90 (s, 1.4H, H_2_-1′), 5.32 (s, 0.6H, H_2_-1′), 6.89–8.06 (m, 8H, H-Ar), 8.31 (s, 0.6H, H-5′), 8.43 (s, 0.4H, H-5′), 10.44 (s, 0.7H, OH), 10.44 (s, 0.3H, OH), 11.84 (s, 0.7H, NH), 12.16 (s, 0.3H, NH). ^13^C NMR (100 MHz, DMSO-*d*_6_): δ (ppm) = 13.2, 23.6, 23.7, 44.7, 48.6, 102.9, 110.5, 110.9, 118.5, 121.3, 122.5, 126.6, 127.7, 129.1, 130.1, 130.9, 133.6, 138.3, 139.7, 145.9, 150.9, 155.7, 156.3, 157.8, 160.0, 165.4, 168.1.


*(E)-N’-(4-bromobenzylidene)-2-(3,6-dimethyl-4-oxo-1-phenyl-1,4-dihydro-5H-pyrazolo[3,4-d]pyrimidin-5-yl)acetohydrazide*
**5h**


Yield: 72%, mp: 248–250 °C. ES-HRMS [M+H]^+^ calcd. for (C_22_H_20_BrN_6_O_2_)^+^: 479.0826, found: 479.0821. ^1^H NMR (400 MHz, DMSO-*d*_6_, mixture of isomers): δ (ppm) = 2.52 (s, 3H, H_3_-12), 2.56 (s, 2.3H, H_3_-13), 2.59 (s, 0.7H, H_3_-13), 4.89 (s, 0.4H, H_2_-1′), 5.32 (s, 1.6H, H_2_-1′), 7.37–8.06 (m, 9H, H-Ar), 8.06 (s, 0.8H, H-5′), 8.22(s, 0.2H, H-5′), 11.94 (s, 1H, NH). ^13^C NMR (100 MHz, DMSO-*d*_6_): δ (ppm) = 13.2, 23.6, 23.7, 44.6, 45.1, 102.9, 121.3, 123.36, 123.45, 126.6, 128.8, 129.0, 129.2, 131.8, 133.2, 138.2, 143.3, 125.9, 146.1, 150.9, 157.8, 159.9, 163.5, 168.3.

### 4.3. Cell Viability Assays

Real time monitoring of cell viability was performed using the Real-Time Cell Analyzer (RTCA) xCELLigence System (ACEA Biosciences, San Diego, CA, USA), as previously reported [67]. MCF-7 and NHDF cells were cultured in Dulbecco’s Modified Eagle’s medium (DMEM) high glucose (4.5 g/L) medium, supplemented with 10% fetal bovine serum, penicillin–streptomycin (100 U/mL), and 2 mM L-glutamine. MCF-7 and NHDF cells were seeded at a cell density of 3000 cells/well and 2000 cells/well, respectively. During the exponential growth phase, approximately 24 h after seeding, medium was removed, and cells were treated with medium containing the synthetic compounds at the indicated concentrations. All compounds were dissolved in DMSO to prepare 20- or 10-mM stock solutions and further diluted in culture medium to perform antiproliferative assays. In all experiments, the final concentration of DMSO did not exceed 0.5%, which was shown to be well tolerated with no obvious toxic effects to cells.

Growth inhibitory effects of synthetic compounds are reported as cell index values relative to controls treated with 0.5% DMSO vehicle. Cell index values were normalized just after the start of treatment to have normalized cell index values (NCI). Growth curves were generated by measuring normalized cell index variations. NCI values and real-time NCI traces were obtained through the RTCA-integrated software (version 2.0.0.1301, ACEA Biosciences, San Diego, CA, USA).

Dose-response curves for **5a**, **5e**, **5g**, and **5h** were modeled using the variable slope sigmoid Hill equation, to determine IC_50_ values against MCF7 and NHDF cells after 24-h and 48-h drug treatments, respectively. As NHDF have longer doubling time than MCF7 cells (42.6 h ± 1.7 versus 21.3 h ± 0.2), cytotoxicity in fibroblasts was evaluated within a wider time window, thereby allowing cells to complete at least one full cell cycle. Dose–response curves and IC_50_ values were calculated using the GraphPad Prism Software Version 5 (GraphPad Software Inc., San Diego, CA, USA).

### 4.4. Apoptosis Assay and Cell Cycle Analysis

After 24 h exposure to IC_50_ concentrations of **5a**, **5e**, **5g**, and **5h**, MCF-7 cells were collected and stained with annexin-V-fluorescein isothiocyanate (FITC) and propidium iodide, using the FITC Annexin V Apoptosis Detection kit I (Becton Dickinson, BD, Franklin, NJ, USA) for detection of apoptotic and necrotic cells by flow cytometry, as previously reported [68]. Samples were prepared according to manufacturer’s protocol.

For analysis of cell cycle distribution, MCF-7 cells (a) were incubated for 24 h with **5a**, **5e**, **5g**, and **5h** at their IC_50_ concentrations, (b) permeabilized with 70% cold ethanol for 1 h, and (c) stained for 30 min with a calcium and magnesium-free PBS solution containing 50 µg/mL propidium iodide (Sigma Aldrich, St. Louis, MO, USA) and 10 µg/mL RNase A (EuroClone S.p.a., Pero, MI, Italy).

Flow cytometry analyses were performed by using a NAVIOS flow cytometer and acquired data elaborated by Kaluza software (Beckman Coulter, Brea, CA, USA). Ten thousand events were acquired for each analyzed sample, and at least three independent experiments were carried out.

### 4.5. Molecular Docking Procedure

The optimization of all the geometries of scaffolds was performed with Gaussian 09 semi-empirical PM3 force-field method [69]. The co-crystal structure of ‘erlotinib’ with EGFR of PDB (PDB: 1M17) was obtained from the RSCB protein data bank. [70] Docking studies were performed using Autodock 4.2 software [71]. The visualization and analysis of interactions were performed using Pymol, version 0.99 [72].

### 4.6. Physicochemical Studies

#### 4.6.1. Synthesis of Oligomers

Oligonucleotides have been synthesized on an ABI 394 DNA/RNA synthesizer (Applied Biosystem, Inc., Waltham, MA, USA) by using the standard DNA synthesis protocol on solid phase at the 5-μmol scale [73]. In particular, the following DNA sequences have been synthesized: d(AGGGCGGTGTGGGAATAGGGAA) (*KRAS* 22RT), d(TAGGGTTAGGGTTAGGGTTAGGG) (Tel23), d(GGGCGCGGGAGGAATTGGGCGGG) (BCL2-G4), and d(CGAATTCGTTTTCGAATTCG) (hairpin duplex). After synthesis, the oligonucleotides were detached from the support and deprotected by treatment with concentrated aqueous ammonia at 55 °C for 12 h. The combined filtrates and washings were concentrated under reduced pressure, dissolved in water, and purified by high-performance liquid chromatography (HPLC) on a Nucleogel SAX column (Macherey-Nagel, 1000–8/46), using buffer A consisting of 20 mM KH_2_PO4/K_2_HPO_4_ aqueous solution (pH 7.0), containing 20% (*v*/*v*) CH_3_CN, buffer B consisting of 1 M KCl, 20 mM KH_2_PO_4_/K_2_HPO_4_ aqueous solution (pH 7.0), containing 20% (*v*/*v*) CH_3_CN, and a linear gradient from 0% to 100% B for 30 min with a flow rate of 1 mL min^−1^. The isolated oligomers were further desalted by Sep-pak cartridges C-18(Waters) and lyophilized. DNA samples were then dissolved in 20 mM phosphate buffer (pH 7.0) containing 60 mM KCl and 0.1 mM EDTA. The concentration of oligonucleotides was determined by UV adsorption measurements at 90 °C using molar extinction coefficient values ε (λ = 260 nm) calculated by the nearest neighbor model [74]. All samples were heated at 90 °C for 5 min, gradually cooled to room temperature overnight, and finally incubated at 4 °C before data acquisition.

#### 4.6.2. Circular Dichroism (CD) Experiments

Circular dichroism (CD) experiments were performed on a Jasco J-815 spectropolarimeter (JASCO Inc., Tokyo, Japan) equipped with a PTC-423S/15 Peltier temperature controller. All the spectra were recorded at 20 °C in the wavelength range of 230−340 nm and averaged over three scans. The scan rate was 100 nm min^−1^, with a 4 s response and 1 nm bandwidth. Buffer baseline was subtracted from each spectrum. A 2 mM oligonucleotide concentration was used. CD spectra of DNA/ligand mixtures were obtained by adding 10 mol equiv of ligands (stock solutions of ligands were 10 mM in DMSO). The CD melting curves were obtained by following changes of CD signal at the wavelength of maximum intensity for the G4 sequences and at the wavelength of minimum intensity for the hairpin duplex (264 nm for *KRAS* 22RT and *BCL2*-G4, 287 nm for Tel23, 251 nm for hairpin duplex). The experiments were carried out in the 20–100 °C range at 1 °C min−1 heating rate and were recorded in the absence and presence of ligands (10 mol equiv) added to the folded DNA structures. The melting temperatures (T_m_) were determined from curve fit using Origin 7.0 software. Δ*T*_m_ values were determined as the difference in the melting temperature of DNA structures with and without ligands. Each experiment was performed in duplicate, and the reported values averaged.

## Data Availability

Not applicable.

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
