# Peer review of "Molecular Docking and Biophysical Studies for Antiproliferative Assessment of Synthetic Pyrazolo-Pyrimidinones Tethered with Hydrazide-Hydrazones"

_ijms, 2021, doi:10.3390/ijms22052742_

Round 1

Reviewer 1 Report

In their study Horchani et al. have synthesised pyrazolo-pyrimidinones tethered with hydrazide-hydrazones and tested their capacity to reduce cell viability of MCF-7 cells. The authors found that some compounds induce apoptosis and lead to cell arrest in the G1/G0 phase.  Moreover, molecular docking simulations suggest high binding between some compounds (5a, 5e, 5g, 5h) and EGFR.

This manuscript reports some interesting data. It can be improved after the following points are addressed.

  1. First, the way the citations are numbered is quite unusual. In the reference list the authors use the decimal numerals, in the text they use the Roman numerals. Please, replace the Roman numerals with the decimal numerals in the text;
  2. Cell viability experiments have been done only in one cancer cell line (MCF-7). Have the authors tested the effect of the compounds also in other cancer cell lines? Is the reported cytotoxicity specific of MCF-7 cells? Can it be observed also in other cells?
  3. As target of the compounds the authors focused on EGFR. What is the rationale of having concentrating on this receptors? Is EGFR overexpressed in MCF-7? If it is not, why selecting MCF-7 cells?
  4. To demonstrate that the designed compounds bind to EGFR, the authors carried out docking experiments. Although these experiments are interesting, they do not provide a direct evidence that the compounds bind to the receptor. The paper would greatly improve if the authors could do experiments, for instance SPR, to demonstrate that there is binding between the designed compounds and EGFR. As far as I know recombinant EGFR is commercially available.

Reviewer 2 Report

Minor considerations

  1. P6, second paragraph. compound 5h, which resulted to be approximately 2 times more potent than”… cisplatin, one of the most widely used and effective chemotherapeutic drugs.” Cisplatin is widely used against ovarian, testicle and bladder cancer. Does this comparison apply for the breast cancer? Please extend a bit this claim.
  2. As related to my above comment. It is important to give insights about the chemicals and their structural characteristics that might lead to the development of more potent compounds. Please include a paragraph (in section Discussion) discussing the comparative differences between the Table in Figure 2B and the Figure 1S. Why ciplatin has better IC50 in one case and worst in another compared to the synthesized compounds.
  3. Please indicate in section 2.3.1 whether the EGFR protein has been pre-treated and how? Were the waters (if any) completely removed or some cavity waters were held before docking?
  4. Introduction, p2-3. Please fix all reference numbers according to the journal requirement. Also p.10. They are in roman numerals.
  5. P2, last paragraph, Replace “molecules D and E (Figure 1)” with “molecules D and E in Figure 1”
  6. Include a relevant reference at the end of the first paragraph in p3 addressing the anticancer drugs related to G-q DNA.
  7. Last sentence in p3. What is the most challenging topic for search of G4 ligands as anticancer drugs?

Round 2

Reviewer 1 Report

The authors have adequately addressed the points raised by this referee in his first round of review. I therefore suggest publication of the manuscript in IJMS